# Rethinking Image Restoration for Object Detection

**Shangquan Sun**
State Key Laboratory of Information Security
Institute of Information Engineering
Chinese Academy of Sciences
Beijing, China 100093
`sunshangquan@iie.ac.cn`

**Wenqi Ren** *
School of Cyber Science and Technology
Sun Yat-sen University, Shenzhen Campus
Shenzhen, China 518107
`rwq.renwenqi@gmail.com`

**Tao Wang**
Huawei Technologies Co., Ltd.
Beijing, China 100085
`wangtao10@huawei.com`

**Xiaochun Cao**
School of Cyber Science and Technology
Sun Yat-sen University, Shenzhen Campus
Shenzhen, China 518107
`caoxiaochun@mail.sysu.edu.cn`

## Abstract

Although image restoration has achieved significant progress, its potential to assist object detectors in adverse imaging conditions lacks enough attention in the research community. It is reported that the existing image restoration methods cannot improve the object detector performance and sometimes even reduce the detection performance. To address the issue, we propose a targeted adversarial attack in the restoration procedure to boost object detection performance after restoration. Specifically, we present an ADAM-like adversarial attack to generate pseudo ground truth for restoration fine-tuning. Resultant restored images are close to original sharp images, and at the same time, lead to better object detection results. We conduct extensive experiments in image dehazing and low light enhancement and show the superiority of our method over conventional training and other domain adaptation and multi-task methods. The proposed pipeline can be applied to all restoration methods and both one- and two-stage detectors.

## 1 Introduction

Image quality degradation is common in outdoor captured images and many image restoration methods have been developed to automatically recover sharp details from the degraded images. Typically, the goal of image restoration is to generate sharper results from which human vision can better identify the structural details and objects in the images. However, in many applications, large-scale images are collected for high-level computer vision tasks (*e.g.*, object detection, semantic segmentation, and autonomous driving) other than a visual examination by human. Though significant progress has been made in terms of the visual quality, its ability to serve as a pre-processing step for downstream vision tasks lacks enough research attention.

Object detection has achieved remarkable success as the boosting of deep neural networks. Many successful models, *e.g.*, Faster-RCNN [32] and YOLOv3 [31], have been developed and deployed in real-world applications such as autonomous driving [4, 38]. However, in the settings of the commonly used benchmark for object detection, poor weather and imaging conditions are intentionally avoided [7] or insufficient due to data bias [16, 9]. The unbalanced data distribution toward clean images leads to a domain shift between properly captured images and low-quality ones in

---

*Corresponding Author.

36th Conference on Neural Information Processing Systems (NeurIPS 2022).

adverse environments [36]. The performance of object detector trained on high-quality samples thus downgrades significantly in adverse imaging and weather conditions such as haze, rain, and low light exposure [23, 5] since the weather-related deterioration features distort captured image [29, 47].

Many existing solutions to the issue lie in domain adaptation from clean images (source domain) to corrupted ones (target domain) [5, 13, 36]. They assume there exists a distribution shift between clean images and those in poor conditions, and adapt detectors by unsupervised weather-specific priors or domain adaptation components to align the features of two distributions to reduce domain discrepancy. However, the detector after adaptation is found to perform worse in the source domain than that without adaptation [23] since adapted detectors may focus on weather-relevant deterioration information. Another track to tackle the issue is to jointly train a simple visual quality enhancement module together with detection network [5, 23]. However, these modules cannot provide satisfactory results compared to powerful restoration methods. Many artifacts and noise patterns occur when applying the enhancement module. It is also challenging to balance visual quality and detection performance by tuning hyper-parameters during training. Since the detector is trained on corrupted samples or the combination of corrupted and clean samples, its performance on clean samples may decrease, similar to the case of domain adaptation.

Different from previous works modifying object detectors, we address the issue by adapting restoration algorithms such that recovered images can achieve both good visual quality and better detection results for detectors. Though image restoration methods produce satisfying results from the perspective of human eyes, detectors may consider them as variables following the distribution of an artificial neural network rather than the real-world, thus being confused by such a covariate shift. Therefore, finding a pseudo ground truth close to the original sharp image potentially improves detection. To find such pseudo ground truth, a targeted adversarial attack can be employed to add an invisibly small perturbation to the original ground truth, on which the detector will give our desired prediction. The resultant adversarial examples are obtained by minimizing the loss of object detectors over perturbation, and can serve as pseudo ground truth for restoration algorithms. Such perturbation is trivial, so restoration models are not impacted to the greatest extent. During the fine-tuning stage, only the weights of restoration methods are updated and detectors keep unchanged. Thus the performance of detectors will not decline in the clean domain. Note that the goal of this paper is not the development of a new image restoration algorithm or a new object detection model. Instead, we propose a targeted attack preprocessing of image ground truth to help improve the accuracy of downstream object detection. Specifically, we propose a momentum-based Adam [17]-variant adversarial example generation method to estimate pseudo ground truth.

Our contributions can be summarized as following:

1. We propose a fine-tuning pipeline to improve detection performance and retain visually satisfactory restoration results. Specifically, we show multi-task optimization problem of balancing both detection and restoration can be decomposed into a targeted adversarial attack generation and a restoration problem. The pseudo ground truth generated by adversarial example generation is used for fine-tuning.

2. We design a momentum-based ADAM-like iterative targeted adversarial example generation algorithm to estimate pseudo ground truth for restoration models fine-tuning. Rather than updating samples by first-order gradient in existing work, we compute ADAM-variant clipped and signed gradient with momentum estimation.

3. We conduct extensive experiments in two image restoration tasks, *i.e.*, dehazing and low light enhancement for both one- and two-stage object detectors. We first evaluate the effect of our adversarial attack algorithm compared to existing ones on test sets. We then show our proposed fine-tuning pipeline effectively improves the detection performance gain of restoration algorithms as a pre-processing step of detectors compared to conventional training and other existing works.

## 2 Related Works

### 2.1 Restoration for Detection

In the image restoration task, many works have achieved promising performance in terms of visual quality and quantitative evaluations [24, 8, 20, 35, 33]. Though many of them mention their potential

usage as a pre-processing for downstream high-level tasks (*e.g.*, object detection and segmentation), only a few of them experiments on how much gain of detection accuracy can be acquired before and after the dehazing step. Dong *et al.* [8] simulate haze on KITTI [10] and show their method removes the synthetic haze effectively and can improve YOLOv3 [31] detection predictions. Li *et al.* [20] publicize the RESIDE dataset, which contains a subset in real-world hazy environment for evaluating object detection on dehazed results. A recent work [23] appends a differentiable image processing module consisting of multiple traditional filters on YOLOv3, and trains the module and YOLOv3 together on the combination of clean and hazy samples. Due to the limitation of filters, it generates many noises and artifacts and thus returns many false positive detection predictions.

In low-light environments, several object detection and semantic segmentation datasets have been collected. Yang *et al.* [46] create a benchmark for face detection in real-world low light condition. Loh *et al.* [26] publish a object detection benchmark in low light environment. Recently, Sakaridis *et al.* [34] collect a comprehensive semantic segmentation dataset in dark imaging condition. To tackle low-light enhancement task, ZeroDCE [12] is developed by formulating light enhancement as a task of image-specific curve estimation in the pixel-wise range. Its performance of helping face detection in low-light conditions is tested on DarkFace [46]. Ma *et al.* [27] recently develop a new Self-Calibrated Illumination (SCI) learning framework for fast, flexible, and robust brightening images in real-world low-light scenarios. They show its improvement for low-light face detection and segmentation tasks on ACDC [34] and DarkFace [46] datasets.

The above-mentioned works either merely experiment on synthetic datasets or have to tune detection models together with a restoration module, which limits the generalizability and extensibility of their works on real-world benchmarks and various tasks. In this work, we focus on proposing a plug-and-play targeted attack module for restoration algorithms to improve the performance of object detection without tuning detection models.

## 2.2 Adversarial Attack on Detector

After Szegedy *et al.* [37] finds that adding an imperceptible perturbation to an image may fool neural network classifier, many adversarial attack generation methods are presented in classification [11, 30, 28, 3, 18]. Fewer works about adversarial attack on object detector are proposed [25] because of its more complicated architecture and loss terms [41, 2]. Some works succeed in fooling detectors by either wearable patch [40, 45, 42] or non-planar patch [50, 48, 43, 14, 39]. Only limited works are about targeted attack. Dense Adversary Generation (DAG) [44], Robust Adversarial Perturbation (RAP) [22], and Contextual Adversarial Perturbation (CAP) [49] are designed for the two-stage detectors consisting of region proposal network. Other works such as Unified and Efficient Adversary (UEA) [41] and Targeted adversarial Objectness Gradient attack (TOG) [6] can attack both one-stage and two-stage detectors with more generic settings. UEA is a Generative Adversarial Network framework trained with class loss and feature loss that generates transferable adversarial images and videos. TOG [6] consists of three relatively powerful adversarial attack algorithms for object-vanishing, object-fabrication, and object-mislabeling attacks, respectively. A recent work [2] develops a content-aware blackbox adversarial attack framework based on TOG. In our work, we use TOG as the baseline for whitebox targeted adversarial example generation due to its flexibility and effectiveness in conducting extensive experiments. We assume detectors are whitebox because our goal is to improve its performance rather than fool it.

## 3 Proposed Method

### 3.1 Pseudo Ground Truth Generation for Restoration

Denote a restoration model with parameters $\omega$ as $\mathcal{R}$ and a detector with parameter $\theta$ as $\mathcal{D}$. Suppose a dataset for training contains three components: corrupted images $\mathbf{x}$, corresponding clean images $\hat{\mathbf{x}}$, and detection annotations $\hat{\mathbf{y}}$. Given that $\mathcal{D}$ only learns on clean samples $\{\hat{\mathbf{x}}, \hat{\mathbf{y}}\}$, the prediction of the detector on corrupted samples $\mathcal{D}(\mathbf{x})$ is probably incorrect. Our aim is to fine-tune a restorer $\mathcal{R}$ such that the recovered image is close to clean image, *i.e.*, $\mathcal{R}(\mathbf{x}) \approx \hat{\mathbf{x}}$, and also the detector give correct prediction on the recovered image, *i.e.*, $\mathcal{D}(\mathcal{R}(\mathbf{x})) = \hat{\mathbf{y}}$.

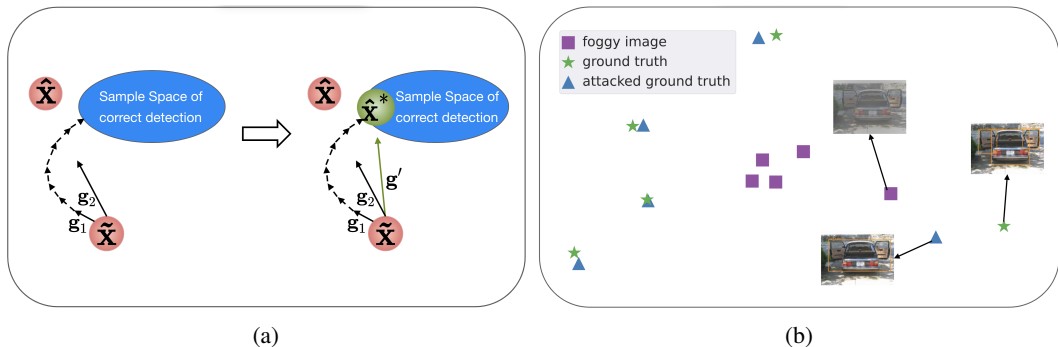

(a)               (b)

Figure 1: 1a A diagram for the case where multi-task training cannot guarantee convergence towards the optimal solution. Suppose the blue elliptical region is a sample space where detectors predict correctly. $\mathbf{g}_1 = \nabla \ell \left( \mathcal{D}(\tilde{\mathbf{x}}), \hat{\mathbf{y}} \right)$ denotes the gradient computed for updating $\theta$, $\mathbf{g}_2$ indicates the gradient of restoration loss, $\|\tilde{\mathbf{x}} - \hat{\mathbf{x}}\|$. Any combination of the two at this iteration cannot make the recovered image $\tilde{\mathbf{x}}$ approach the elliptical region. However, we can find a pseudo ground truth in the region, $\hat{\mathbf{x}}^*$, that is as close to $\hat{\mathbf{x}}$ as possible. Updating towards it satisfy both the constraint of restoration and the requirement of correct detection. 1b is VGG16 features distribution of five *car* patches. Each point represents a image patch. ★ is clean image, ■ denotes corrupted patches, and ▲ is attacked pseudo ground truth. We can clearly find that a feature shift occurs as its detection result gets better.

Traditional supervised restoration problem is not involved with the detection objective and can be generally formulated as:

$$\tilde{\mathbf{x}} = \arg\min_{\mathcal{R}} \|\mathcal{R}(\mathbf{x}) - \hat{\mathbf{x}}\|. \tag{1}$$

However, solely training with the objective 1 cannot guarantee a satisfactory result in subsequent object detection since the covariate shift and artifacts as reported in [20, 23, 21]. Given that restoration is an ill-posed problem, various possible recovered images exist for a degraded one. Considering the inevitable image noises due to hardware limitations during the image acquisition and transmission stage especially in adverse weather and imaging conditions, captured clean images can unnecessarily be regarded as a perfect ground truth.

Unlike conventional restoration methods, our goal is to find a balance between visual quality and detection performance on restored images. Our objective is to make detector $\mathcal{D}$ inference correctly on the restored image as close as possible to the clean image. Thus it can be considered as a multi-task optimization problem by combining the restoration constraint and object detection loss. We start by formulating it as follows,

$$\begin{aligned} p_1 &:= \min_{\mathcal{R}} \ell \left( \mathcal{D}(\mathcal{R}(\mathbf{x})), \hat{\mathbf{y}} \right), \text{ s.t. } \|\hat{\mathbf{x}} - \mathcal{R}(\mathbf{x})\| \leq s \\ &= \min_{\tilde{\mathbf{x}}} \ell \left( \mathcal{D}(\tilde{\mathbf{x}}), \hat{\mathbf{y}} \right), \text{ s.t. } \|\hat{\mathbf{x}} - \tilde{\mathbf{x}}\| \leq s, \end{aligned} \tag{2}$$

where $\tilde{\mathbf{x}} = \mathcal{R}(\mathbf{x})$ is the recovered image, $s$ is a slackness coefficient indicating the allowed distance from the ground truth $\hat{\mathbf{x}}$. The constraint optimization problem cannot be solved easily due to the constraint. Thus we formulate its dual function as below by introducing a dual variable $\alpha \geq 0$ as,

$$d_1 := \min_{\tilde{\mathbf{x}}} \mathcal{L}(\tilde{\mathbf{x}}) = \ell \left( \mathcal{D}(\tilde{\mathbf{x}}), \hat{\mathbf{y}} \right) + \alpha \|\hat{\mathbf{x}} - \tilde{\mathbf{x}}\|. \tag{3}$$

**Theorem 1** *For any $s > 0$, there exists $\alpha \geq 0$ such that the optimal solution of (2), $\tilde{\mathbf{x}}^*$, is also optimal for* (3)*, and vise versa.*

The proof of Theorem 1 is attached in the supplemental materials. The main idea is to use Lagrangian and a strong duality assumption. By Theorem 1, we can say the original problem Eq.2 and the dual problem is equivalent. However, the multi-task training faces difficulty in tuning the hyper-parameter $\alpha$, and an improper $\alpha$ leads to either poor detection or restoration results. Figure 1 shows a case where optimizing $d_1$ cannot ensure correct detection.

Thus we further reformulate the objective and bypass the issue. By substituting $\beta = \frac{1}{\alpha}$, we get

$$d_2 := \min_{\tilde{\mathbf{x}}} \|\hat{\mathbf{x}} - \tilde{\mathbf{x}}\| + \beta \ell \left( \mathcal{D}(\tilde{\mathbf{x}}), \hat{\mathbf{y}} \right). \tag{4}$$

**Algorithm 1:** Framework of Adam-variant adversarial example generation.

---

**Input:** Input dataset $\mathcal{U}$ with samples $\mathbf{x}$ with annotation $\hat{\mathbf{y}}$, Detector $\mathcal{D}$, Number of attack iteration $T$, Update stepsize $\lambda$, Magnitude tolerance of Perturbation $\delta$
**Output:** Attacked adversarial examples $\hat{\mathbf{x}}^*$,

---

Assign $\beta_1 = 0.9$, $\beta_2 = 0.999$, $\epsilon = 1e - 8$
Initialize $\mathbf{m}_0 = 0$, $\mathbf{v}_0 = 0$, $\mathbf{z} = rand(-\delta, \delta)$;
$\mathbf{x} \leftarrow \text{clip}(\mathbf{x} + \mathbf{z}, 0, 1)$
**foreach** *($\mathbf{x}$, $\hat{\mathbf{y}}$) in $\mathcal{U}$* **do**

    **foreach** *$t$ in $1...T$* **do**
        $\mathbf{g_x} \leftarrow \nabla_{\mathbf{x}} \mathcal{L}_\theta(\mathcal{D}(\mathbf{x}), \hat{\mathbf{y}})$
        $\mathbf{m}_t = \beta_1 \mathbf{m}_{t-1} + (1 - \beta_1)\mathbf{g_x}$
        $\mathbf{v}_t \leftarrow \beta_2 \mathbf{v}_{t-1} + (1 - \beta_2)\mathbf{g_x}^2$
        $\hat{\mathbf{m}} \leftarrow \mathbf{m}_t / (1 - \beta_1^t)$
        $\hat{\mathbf{v}} \leftarrow \mathbf{v}_t / (1 - \beta_2^t)$
        Update $\mathbf{x} \leftarrow \mathbf{x} - \lambda \text{sign}[\hat{\mathbf{m}}/(\sqrt{\hat{\mathbf{v}}} + \epsilon)]$;
        $\mathbf{z} \leftarrow \text{clip}(\mathbf{x} - \hat{\mathbf{x}}, -\delta, \delta)$
        $\mathbf{x} \leftarrow \text{clip}(\mathbf{x} + \mathbf{z}, 0, 1)$
    Store $\mathbf{x}$ as pseudo clean image.

---

By Theorem 1 again, we have

$$p_2 := \min_{\tilde{\mathbf{x}}} \|\hat{\mathbf{x}} - \tilde{\mathbf{x}}\|, \text{ s.t. } \ell(\mathcal{D}(\tilde{\mathbf{x}}), \hat{\mathbf{y}}) \leq r, \tag{5}$$

where $r$ is a small slackness coefficient that regularizes the detection performance. By introducing an intermediate variable $\hat{\mathbf{x}}^*$ that is close enough to $\hat{\mathbf{x}}$, we have the expression of optimization,

$$\min_{\tilde{\mathbf{x}}} \|\hat{\mathbf{x}}^* - \tilde{\mathbf{x}}\|, \text{ where } \hat{\mathbf{x}}^* = \arg\min_{\mathbf{x}} \|\mathbf{x} - \hat{\mathbf{x}}\| \text{ s.t. } l(\mathcal{D}(\mathbf{x}), \hat{\mathbf{y}}) \leq r. \tag{6}$$

The solutions of (5) and (6) are equivalent when $\tilde{\mathbf{x}}^* = \hat{\mathbf{x}}^*$. The problem can thus be separated into two optimization steps,

$$\hat{\mathbf{x}}^* = \arg\min_{\mathbf{x}} \|\hat{\mathbf{x}} - \mathbf{x}\|, \text{ s.t. } \mathcal{D}(\mathbf{x}) = \hat{\mathbf{y}}, \tag{7}$$

$$\mathcal{R}^* = \arg\min_{\mathcal{R}} \|\mathcal{R}(\mathbf{x}) - \hat{\mathbf{x}}^*\|. \tag{8}$$

Therefore, (7) is an objective of targeted adversarial attack if we substitute for $\mathbf{x} = \hat{\mathbf{x}} + \mathbf{z}$ and add box constraint $\|\mathbf{x}\| \leq 1$. Then, (7) can yield a detector-friendly sample close to the original ground truth, and (8) forces restoration models to learn the pattern mapping corrupted samples to the pseudo ground truth. The solution of (5) can be approximately computed by the fine-tuning pipeline of sequentially optimizing (7) and (8).

## 3.2 Targeted Adversarial Attack on Object Detector

Object detector $\mathcal{D}$ receives an input image $\mathbf{x}$ and predicts label with $n$ detected objects, *i.e.*, $\mathbf{y} = \{(b^x, b^y, b^H, b^W, C, p)_i | i = 1...n\}$, where $(b^x, b^y, b^H, b^W)$ represents the location and region of bounding box for $i$-th object, $C$ is classification result and $p$ is prediction confidence. The existing method, TOG [6] designs a group of loss terms specific for object detector. The adversarial example is updated iteratively by signed and clipped first-order gradient of the loss term as,

$$\begin{aligned}
\mathbf{x}^{t+1} &\leftarrow \mathbf{x}^t - \lambda \text{sign}\left[\nabla_{\mathbf{x}^t} \mathcal{L}\left(\mathcal{D}(\mathbf{x}^t), \hat{\mathbf{y}}\right)\right] \\
\mathbf{x}^{t+1} &\leftarrow \mathbf{x} + \text{clip}_\delta\left(\mathbf{x}^{t+1} - \mathbf{x}\right).
\end{aligned} \tag{9}$$

where $\lambda$ is the step size of perturbation update and $\delta$ is the allowed perturbation scale centered at original input $\mathbf{x}$. However, we note that the first-order gradient descent used in TOG may get stuck at sub-optimal or converge slowly. Therefore, we introduce the momentum-based optimizer ADAM [17] to optimize adversarial examples during updating perturbation, which is widely used in optimizing parameters of neural networks during training. It can adaptively adjust learning rate

by second-order momentum and overcome sub-optimal with cumulative momentum. Therefore, we reformulate (9) into

$$
\begin{aligned}
\mathbf{g} &\leftarrow \nabla_{\mathbf{x}^t} \mathcal{L}\left(\mathcal{D}(\mathbf{x}^t), \hat{\mathbf{y}}\right), \\
\mathbf{x}^{t+1} &\leftarrow \mathbf{x}^t - \lambda \mathrm{sign}\left[\mathrm{ADAM}(\mathbf{g})\right] \\
\mathbf{x}^{t+1} &\leftarrow \mathbf{x} + \mathrm{clip}_\delta\left(\mathbf{x}^{t+1} - \mathbf{x}\right).
\end{aligned}
\tag{10}
$$

Finally, the generated pseudo ground truth $\hat{\mathbf{x}}^*$ are used to fine-tune restoration models. The main step of the targeted adversarial attack is presented in Algorithm 1. The convergence analysis of the algorithm is discussed in the supplemental material.

## 4 Experiments

### 4.1 Experimental Settings

We conduct experiments on two image restoration tasks, *i.e.*, image dehazing and low-light image enhancement, and two detectors of YOLOv3 [31] and Faster-RCNN [32].

**Dehazing Dataset:** RTTS [20] is a real-world hazy dataset with detection annotation for test purpose. The object labels contain five categories, *i.e.*, *person, car, bus, bicycle, and motorcycle*. To construct a training set with both ground truth clean image and bounding box annotation, we select those PASCAL VOC images with the above-mentioned five classes and simulate haze on them following [23]. We only simulate two scales of haze rather than ten in [23], and thus our training set is only 20% of theirs. RTTS contains 4,322 real-world hazy images. VOC_fog_train and VOC_fog_test consist of 16,222 and 2,734 synthetic hazy images respectively.

**Low-light Enhancement Dataset:** ExDark [26] is a natural low light dataset for object detection consisting of ten classes, *i.e.*, *bicycle, boat, bottle, bus, car, cat, chair, dog, motorbike, person*. Similar to dehazing task, we select those samples in PASCAL VOC with these ten categories and simulate low light corruption to form the training set, dubbed as VOC_dark_train. ExDark contains 2,563 real-world dark images. VOC_dark_train and VOC_dark_test consist of 12,334 and 3,760 synthetic dark images respectively.

**Metrics:** We adopt mean average precision (mAP) with an IOU threshold 50% as a metric for detection performance and Peak Signal-to-Noise Ratio (PSNR) and Structure Similarity Index Measure (SSIM) for restoration. Since our algorithm focuses on improving detection performance without sacrificing visual quality, we expect a gain of mAP and as few degradations in PSNR/SSIM as possible.

**Baselines:** We evaluate our proposed fine-tuning procedure on four restoration models, *i.e.*, MS-BDN [8] and GridDehaze [24] for the dehazing task as well as ZeroDCE [12] and SCI [27] for the low light enhancement task. We compare our fine-tuning pipeline with the traditional learning with only restoration loss. To test the generalization of our pipeline, we conduct experiments for two widely used detectors, *i.e.*, YOLOv3 [31] and Faster-RCNN [32]. We also compare our method against some domain-adaption approaches including DA-YOLO [5], DSNet [15], and IA-YOLO [23].

**Implementation Details** We use YOLOv3 [47] with Darknet-53 backbone. We train both YOLOv3 [31] and Faster-RCNN [32] on the clean images of VOC_fog_train and VOC_dark_train respectively. Then, their parameters are fixed during fine-tuning image restoration methods. No data augmentation is utilized in the experiments. All experiments are conducted on an Nvidia Tesla V100 PCIe 32G GPU and implemented by PyTorch. We fine-tuned models with batch 1 at each iteration without cropping since the adversarial attack on detectors requires the whole image as input. The tuning number of epochs is 10 for all restoration models. The learning rate is set 1e-4 and the optimizer is ADAM with default settings for all models. For better learning for restoration, we replace all L1 loss with Charbonnier loss [1],

$$
\mathcal{L}_{Charbonnier}(\mathbf{x}, \hat{\mathbf{x}}) = \frac{1}{HWC} \sum_{i=1}^{H} \sum_{j=1}^{W} \sum_{k=1}^{C} \sqrt{(\mathbf{x}_{i,j,k} - \hat{\mathbf{x}}_{i,j,k})^2 - \lambda^2},
\tag{11}
$$

where $\lambda$ is a constant for numerical stability. We set $\lambda$ as 1e-6. It is a variant of L1 loss and reported better in restoration [19].

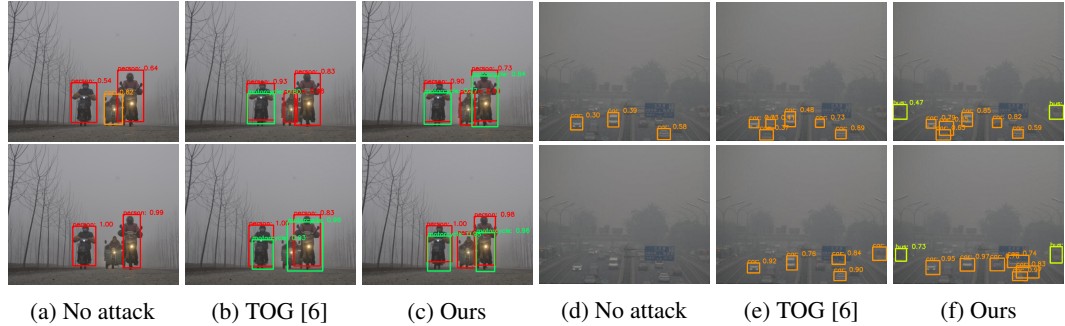

| (a) No attack | (b) TOG [6] | (c) Ours | (d) No attack | (e) TOG [6] | (f) Ours |

Figure 2: A comparison of object detection of different adversarial attack results on RTTS [20]. The 1st row is the result of YOLOv3 [31] and the 2nd is that of Faster-RCNN [32].

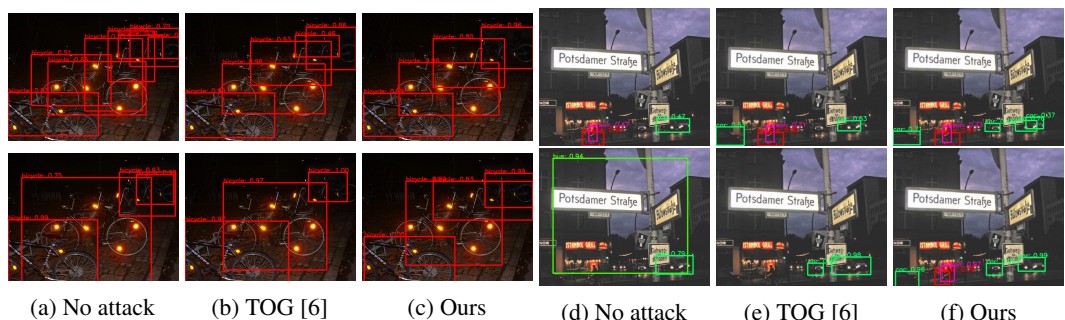

| (a) No attack | (b) TOG [6] | (c) Ours | (d) No attack | (e) TOG [6] | (f) Ours |

Figure 3: A comparison of object detection of different adversarial attack results on ExDark [26]. The 1st row is the result of YOLOv3 [31] and the 2nd is that of Faster-RCNN [32].

## 4.2 Comparisons

### 4.2.1 Targeted Attack on Test Set

We first evaluate whether a pseudo ground truth that enable better detection exists by performing adversarial attack on test sets of RTTS, VOC_fog_test, VOC_dark_test, and ExDark. We compare our method against TOG [6]. The result for YOLOv3 [31] are given in Table 1 and that for Faster-RCNN [32] are reported in the supplemental material. We can observe that both detectors face significant performance degeneration from clean images to corrupted ones. The mAP of YOLOv3 decreases by nearly 15% (15.03% and 14.11%) on degraded images compared to clean ones for both VOC_fog_test and VOC_dark_test. After the targeted attack with a small perturbation, detection accuracy increases significantly. For example, the mAP of YOLOv3 increases from 42.77% on RTTS to over 70% after the attack. Compared to TOG, our algorithm can boost detection performance by 2% - 7% higher.

We show two groups of samples for RTTS in Figure 2 and ExDark in Figure 3. As shown, the detectors cannot work correctly on corrupted inputs. But attacked images enable detectors to predict better and more bounding boxes. Compared to TOG, the images generated by our method lead to more accurate detection predictions. For example, Faster-RCNN cannot detect any car or bus in Figure 2d. However, it can detect 6 cars and 2 buses correctly on the attacked image by our method.

These experiments demonstrate that the theoretical pseudo ground truth that is close to the original ground truth, and enables better detection exists.

### 4.2.2 Dehazing for Detection

Then we show the experiments of restoration by fine-tuning with pseudo ground truth. We train restoration methods on VOC_fog_train and VOC_dark_train and test them on VOC_fog_test, RTTS, VOC_dark_test and ExDark. During fine-tuning with our proposed pipeline, clean images are replaced by attacked ones.

Table 1: The detection performance gain by different targeted adversarial attack methods on YOLOv3 [31]. $\delta = 2/255$ and $\lambda = 1/255$. The experiment results with other configurations and Faster-RCNN is shown in supplemental materials.

| | RTTS | | | Hazy images of VOC_fog_test | | | Clean images of VOC_fog_test | | |
|---|---|---|---|---|---|---|---|---|---|
| | no attack | TOG [6] | Ours | no attack | TOG [6] | Ours | no attack | TOG [6] | Ours |
| bicycle | 40.32 | 66.07 | **74.99** | 61.95 | 91.66 | **93.14** | 82.31 | 95.32 | **95.59** |
| bus | 22.99 | 64.76 | **75.02** | 67.52 | 93.72 | **96.18** | 77.78 | 96.54 | **98.10** |
| car | 49.93 | 73.84 | **79.37** | 75.24 | 94.89 | **96.46** | 83.83 | 96.49 | **97.70** |
| motorcycle | 36.24 | 72.93 | **78.19** | 58.13 | 88.85 | **90.64** | 82.06 | 91.50 | **94.27** |
| person | 64.39 | 82.27 | **84.91** | 71.56 | 91.61 | **94.19** | 83.58 | 94.20 | **95.74** |
| mAP | 42.77 | 71.98 | **78.50** | 66.88 | 92.15 | **94.12** | 81.91 | 94.81 | **96.28** |

| | ExDark | | | Dark images of VOC_dark_test | | | Clean images of VOC_dark_test | | |
|---|---|---|---|---|---|---|---|---|---|
| | no_attack | TOG [6] | Ours | no_attack | TOG [6] | Ours | no_attack | TOG [6] | Ours |
| bicycle | 57.39 | 76.48 | **80.98** | 65.71 | 82.63 | **87.81** | 71.80 | 85.04 | **89.21** |
| boat | 21.15 | 52.03 | **63.42** | 27.58 | 55.16 | **67.77** | 38.44 | 63.38 | **73.62** |
| bottle | 53.85 | 76.41 | **82.05** | 47.98 | 68.99 | **77.33** | 60.37 | 79.91 | **85.25** |
| bus | 63.44 | 86.53 | **89.59** | 67.47 | 82.33 | **89.07** | 82.72 | 92.58 | **94.26** |
| car | 46.63 | 71.37 | **76.99** | 65.74 | 80.21 | **85.91** | 82.36 | 90.78 | **93.17** |
| cat | 46.78 | 79.95 | **83.31** | 66.40 | 87.20 | **91.31** | 82.30 | 93.76 | **96.58** |
| chair | 31.24 | 72.47 | **79.01** | 30.14 | 53.1 | **62.76** | 50.99 | 71.09 | **79.93** |
| dog | 48.94 | 86.70 | **91.14** | 61.36 | 86.56 | **90.62** | 76.20 | 92.98 | **96.18** |
| motorbike | 40.43 | 63.38 | **65.63** | 68.57 | 80.49 | **83.40** | 82.73 | 87.78 | **89.21** |
| person | 52.82 | 77.56 | **80.75** | 67.84 | 80.18 | **83.90** | 81.97 | 89.54 | **90.94** |
| mAP | 46.27 | 74.29 | **79.29** | 56.88 | 75.69 | **81.99** | 70.99 | 84.68 | **88.83** |

Table 2: The quantitative results of existing methods and ours with both restoration metrics and detection metrics on VOC_fog_test and RTTS. CT refers to conventional training.

| | | DAYOLO | DSNet | IA-YOLO | YOLOv3 | YOLOv3+MSBDN | | | YOLOv3+GridDehaze | | |
|---|---|---|---|---|---|---|---|---|---|---|---|
| | | | | | | CT | TOG | Ours | CT | TOG | Ours |
| VOC_fog_test | mAP | 55.11 | 67.40 | 67.40 | 66.88 | 77.06 | 77.52 | 77.66 | 69.29 | 77.93 | 78.36 |
| | PSNR | / | / | 14.65 | 13.50 | 28.72 | 28.54 | 28.53 | 27.83 | 27.41 | 27.42 |
| | SSIM | / | / | 0.5914 | 0.5411 | 0.8852 | 0.8827 | 0.8823 | 0.8726 | 0.8644 | 0.8667 |
| RTTS | mAP | 29.93 | 28.91 | 37.08 | 42.77 | 43.87 | 44.02 | 44.10 | 42.13 | 42.58 | 42.62 |

The main results of dehazing for detection by YOLOv3 [31] are shown in Table 2. We can find that MSBDN [8] fine-tuned by our algorithm can boost detection performance compared to conventional training methods with original ground truth and fine-tuning with TOG attack. Similar tendencies exist for GridDehaze [24]. We show two groups of examples in Figure 2. Our method yields better detection accuracy and keeps adequate restoration performance. However, previous work [23] as shown in Figure 4b outputs unnatural images with many artifacts and noises.

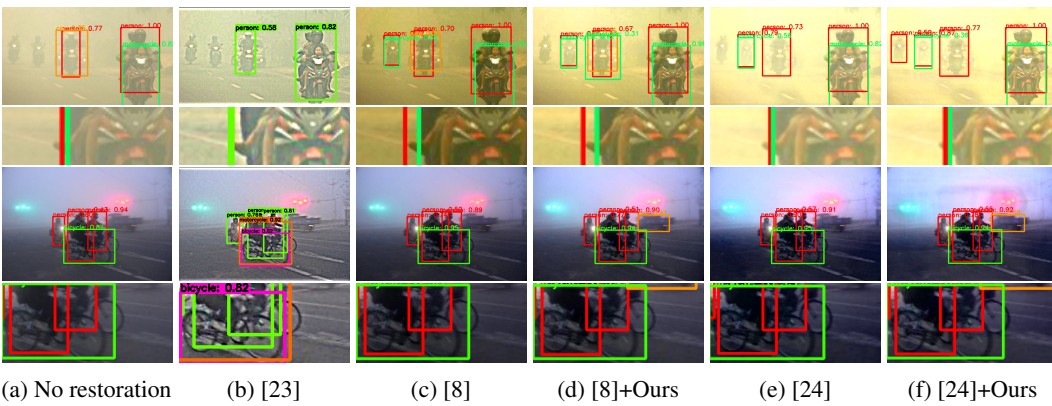

| (a) No restoration | (b) [23] | (c) [8] | (d) [8]+Ours | (e) [24] | (f) [24]+Ours |

Figure 4: A comparison of object detection of different adversarial attack results on RTTS [20]. The last four columns perform detection by YOLOv3[31].

Table 3: The quantitative results of existing methods and ours with both restoration metrics and detection metrics on VOC_dark_test and ExDark. CT refers to conventional training.

| | | DAYOLO | DSNet | IA-YOLO | YOLOv3 | YOLOv3+SCI | | | YOLOv3+ZeroDCE | | |
|---|---|---|---|---|---|---|---|---|---|---|---|
| | | | | | | CT | TOG | Ours | CT | TOG | Ours |
| VOC_dark_test | mAP | 21.53 | 43.75 | 59.40 | 56.88 | 59.54 | 59.68 | 59.87 | 58.21 | 58.43 | 58.49 |
| | PSNR | / | / | 12.43 | 11.99 | 13.44 | 13.45 | 13.96 | 18.23 | 17.90 | 17.87 |
| | SSIM | / | / | 0.5252 | 0.3923 | 0.4891 | 0.4908 | 0.5054 | 0.6350 | 0.6362 | 0.6357 |
| ExDark | mAP | 18.15 | 36.97 | 40.37 | 46.27 | 48.47 | 48.08 | 48.59 | 41.01 | 41.09 | 41.21 |

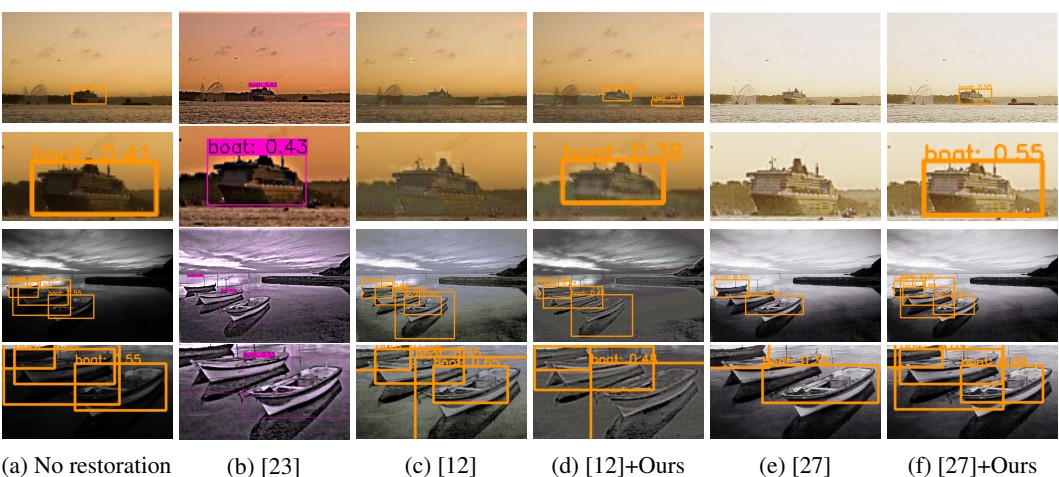

| (a) No restoration | (b) [23] | (c) [12] | (d) [12]+Ours | (e) [27] | (f) [27]+Ours |
|---|---|---|---|---|---|

Figure 5: A comparison of object detection of different adversarial attack results on ExDark [20]. The last four columns perform detection by YOLOv3[31].

### 4.2.3 Low Light Enhancement for Detection

We show the main results of low light enhancement for object detection by YOLOv3 [31] in Table 3. Compared to conventional training with original ground truth and fine-tuning with TOG attacked, SCI [27] fine-tuned by our method boosts detection performance. Similar tendencies exist for ZeroDCE [12]. We show two groups of examples in Figure 3. Our method yields better detection accuracy and, at the same time, keeps fair restoration performance.

### 4.2.4 Limitation

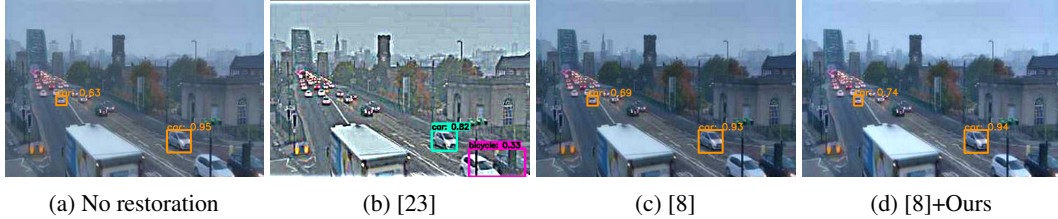

| (a) No restoration | (b) [23] | (c) [8] | (d) [8]+Ours |
|---|---|---|---|

Figure 6: A failure case of object detection of different methods on RTTS [20]. The detector for (a), (c) and (d) is YOLOv3 [31]. (a) is the result without restoration. (c) is the result after the restoration of MSBDN [8] with conventional fine-tuning. (d) is the result after the restoration of MSBDN [8] with our fine-tuning pipeline.

We note that directly performing the adversarial attack on test sets leads to a significant boost in detection results as shown in Table 1. However, the restoration models learning the map from corrupted image to pseudo ground truth cannot yield such a significant improvement. The reason may be the restoration models are incapable of learning perfectly the pattern of attacked pseudo ground truth. The error between original and pseudo ground truths, which makes the attack ineffective. We

show a failure case in Figure 6. Though the image is less hazy after restoration, the detector cannot obtain a better prediction on the recovered image.

## 5   Conclusion

In this work, we present a training procedure for image restoration aimed at better performance for downstream detectors without much visual quality degeneration. The training is supervised by pseudo ground truth generated by the proposed ADAM-like targeted adversarial attack. We conduct extensive experiments in two restoration tasks, *i.e.*, dehazing and low light enhancement for both one- and two-stage object detectors. We show our training pipeline effectively improves the detection performance gain of restoration algorithms as a pre-processing step compared to conventional training and other existing works.

## Acknowledgments and Disclosure of Funding

Supported by the National Key R&D Program of China under Grant 2018AAA0102503, National Natural Science Foundation of China (No. 62025604, 62172409, U1803264), Beijing Natural Science Foundation (No. M22006).

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
