# Rethinking Image Restoration for Object Detection

## Overview

In the supplemental material, we first give the proof of Theorem 1 in Section 1. We then analyze the convergence rate of Algorithm 1 in Section 2. Then we attach more results of targeted attack results on test sets for YOLOv3 [14] and Faster-RCNN [15] in Section 3. We then qualitatively compare our method with conventional training on several more real-world samples from RTTS and ExDark in Section 4. In Section 5 the statistics of object detection datasets are provided.

## 1   Proof of Theorem

**Proposition 1** *Let $f, g : U \to \mathbb{R}$ be convex functions in a domain $U \subset \mathbb{R}^d$. We have two optimization problems with coefficients $s > 0$ and $\beta > 0$*

$$\min_{\mathbf{x} \in U} f(\mathbf{x}) \; s.t. \; g(\mathbf{x}) \le s, \tag{1}$$

$$\min_{\mathbf{x} \in U} f(\mathbf{x}) + \beta g(\mathbf{x}). \tag{2}$$

*We assume that some constraint qualification such as Slater Condition is satisfied for (1). Strong duality thus holds for the above problem. Then for any $\beta > 0$, there exist $s > 0$ and vice versa, such that optimization problems (1) and (2) are equivalent.*

**Proof 1.1 ((1)$\to$(2))** *Suppose $s > 0$ and $\mathbf{x}^*$ is the optimal solution of (1). We have the Lagrangian of (1) with Lagrangian Multiplier $\beta \ge 0$*

$$\mathcal{L}(\beta, \mathbf{x}) = f(\mathbf{x}) + \beta(g(\mathbf{x}) - s) \tag{3}$$

*By the difinition of Lagrangian dual problem, $\beta^*$ is optimal for*

$$\max_{\beta \ge 0} \min_{\mathbf{x}} \mathcal{L}(\beta, \mathbf{x}) = f(\mathbf{x}) + \beta(g(\mathbf{x}) - s) \tag{4}$$

*The assumption of strong duality gives rise to*

$$\max_{\beta \ge 0} \min_{\mathbf{x}} \mathcal{L}(\beta, \mathbf{x}) = \min_{\mathbf{x}} \max_{\beta \ge 0} \mathcal{L}(\beta, \mathbf{x}) \tag{5}$$

16  $\mathbf{x}^*$ *is the optimal solution of the saddle point problem when $\beta$ reaches optimal*

$$\begin{aligned}
\mathbf{x}^* &= \arg\min_{\mathbf{x}} \max_{\beta \geq 0} \mathcal{L}(\beta, \mathbf{x}) = f(\mathbf{x}) + \beta(g(\mathbf{x}) - s) \\
&= \arg\min_{\mathbf{x}} \mathcal{L}(\mathbf{x}) = f(\mathbf{x}) + \beta^*(g(\mathbf{x}) - s) \\
&= \arg\min_{\mathbf{x}} \mathcal{L}(\mathbf{x}) = f(\mathbf{x}) + \beta^* g(\mathbf{x})
\end{aligned} \tag{6}$$

17  *Therefore, $\mathbf{x}^*$ is optimal for both (1) and (2) when $\beta = \beta^*$.*

18  **Proof 1.2 ((2)→(1))** *Suppose $\beta > 0$ and $\mathbf{x}^*$ is the optimal solution of (2). We want to show $\mathbf{x}^*$ is*
19  *also optimal for (1). Let $s = g(\mathbf{x}^*)$. If there exist an optimal solution $\hat{\mathbf{x}} \neq \mathbf{x}^*$ for (1) such that*
20  *$g(\hat{\mathbf{x}}) \leq s$, we have*

$$\begin{aligned}
& f(\hat{\mathbf{x}}) < f(\mathbf{x}^*) \\
\Rightarrow\ & f(\hat{\mathbf{x}}) + \beta g(\hat{\mathbf{x}}) < f(\mathbf{x}^*) + \beta s \\
\Rightarrow\ & f(\hat{\mathbf{x}}) + \beta g(\hat{\mathbf{x}}) < f(\mathbf{x}^*) + \beta g(\mathbf{x}^*),
\end{aligned} \tag{7}$$

21  *which contradicts that $\mathbf{x}^*$ is optimal for (2).*

## 2  Convergence Analysis

23  The proof of ADAM in the original paper [7] is found incomplete by several works [18, 16]. A
24  failure case of ADAM is found in [16], caused by the exponential moving average. Bock *et al.* [1]
25  prove the local convergence in batch mode on a fixed training set. In our case, since we are
26  optimizing over image rather than network parameters, the assumption of deterministic training set
27  holds. Ward *et al.* [17] show the standard convergence rate of ADAM for a non-convex problem
28  is $O(\ln(N)/\sqrt{N})$ with a scalar stepsize. Zou *et al.* [18] show that the sufficient conditions of
29  ADAM's convergence are an appropriate initial learning rate $1/\sqrt{N}$ and exponential moving average
30  scale $\beta_2 = 1 - 1/N$, given $N$ the number of steps. Défossez *et al.* [3] give a simplified proof
31  leading to the same convergence rate and conditions and extend the best known bound of ADAM
32  from $O((1 - \beta_1)^{-5})$ to $O((1 - \beta_1)^{-1})$. The sign function used in previous works of adversarial
33  attacks [8, 5] does not affect the convergence if we consider it as a fixed updating rate $\lambda$. The
34  clamping operation restricting perturbation scale $\delta$ and box constraint within $[0, 1]$ may confine the
35  convergence but it is necessary for optimization settings.

## 3 More attack results

We show more results of attack on test sets in the section. In Table 1, the detection performance gain of different attack methods for Faster-RCNN [15] is given. We can find that our method shows higher mAP boost than TOG [2]. We further give more visualization results of detection in Figure 1, 2 for hazy dataset RTTS [9] and Figure 3, 4 for low ligh dataset ExDark [12].

Table 1: The detection performance gain by different targeted adversarial attack methods on Faster-RCNN [15]. $\delta = 2/255$ and $\lambda = 1/255$.

| | RTTS | | | VOC_fog_test | | | VOC_clean_test | | |
|---|---|---|---|---|---|---|---|---|---|
| | no attack | TOG [2] | Ours | no attack | TOG [2] | Ours | no attack | TOG [2] | Ours |
| bicycle | 27.15 | 48.38 | **55.65** | 44.82 | 82.46 | **86.49** | 76.34 | 87.78 | **91.22** |
| bus | 12.70 | 38.76 | **48.26** | 54.37 | 91.37 | **94.59** | 82.24 | 95.52 | **97.45** |
| car | 31.29 | 44.46 | **47.52** | 61.32 | 85.79 | **88.99** | 81.83 | 91.00 | **93.29** |
| motorcycle | 16.90 | 46.28 | **50.14** | 36.28 | 80.92 | **87.27** | 73.47 | 87.03 | **90.77** |
| person | 58.55 | 67.24 | **70.31** | 52.24 | 83.16 | **86.03** | 76.58 | 86.33 | **88.97** |
| mAP | 29.32 | 49.02 | **54.37** | 49.81 | 84.74 | **88.67** | 78.09 | 89.53 | **92.34** |

| | ExDark | | | VOC_dark_test | | | VOC_clean_test | | |
|---|---|---|---|---|---|---|---|---|---|
| | no_attack | TOG [2] | Ours | no_attack | TOG [2] | Ours | no_attack | TOG [2] | Ours |
| bicycle | 44.71 | 73.09 | **78.72** | 57.20 | 79.23 | **83.1** | 76.61 | 83.83 | **87.79** |
| boat | 31.79 | 66.8 | **75.68** | 49.45 | 75.90 | **83.58** | 62.98 | 82.58 | **86.74** |
| bottle | 42.29 | 64.17 | **70.56** | 38.95 | 61.92 | **68.94** | 52.41 | 69.83 | **75.76** |
| bus | 50.65 | 87.59 | **88.68** | 64.53 | 85.11 | **92.01** | 78.62 | 90.82 | **94.07** |
| car | 38.58 | 62.71 | **68.94** | 67.48 | 82.27 | **86.91** | 81.71 | 88.80 | **90.96** |
| cat | 41.38 | 78.97 | **83.74** | 68.72 | 87.83 | **92.31** | 86.08 | 94.83 | **97.74** |
| chair | 35.06 | 77.14 | **82.62** | 29.84 | 64.23 | **74.90** | 49.94 | 74.64 | **84.14** |
| dog | 46.70 | 85.98 | **91.01** | 61.34 | 90.01 | **95.05** | 80.43 | 93.73 | **97.81** |
| motorbike | 27.76 | 54.55 | **62.76** | 65.04 | 79.93 | **86.52** | 75.71 | 85.93 | **89.30** |
| person | 40.21 | 62.60 | **68.68** | 61.14 | 75.87 | **81.28** | 77.48 | 84.19 | **86.64** |
| mAP | 39.91 | 71.36 | **77.14** | 56.37 | 78.23 | **84.46** | 72.20 | 84.92 | **89.09** |

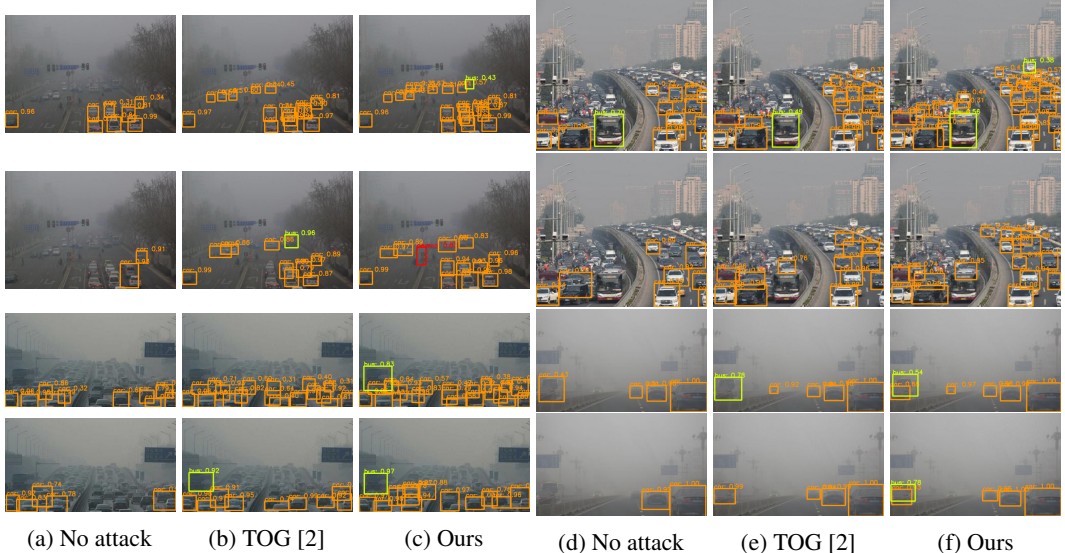

(a) No attack     (b) TOG [2]     (c) Ours     (d) No attack     (e) TOG [2]     (f) Ours

Figure 1: A comparison of object detection of different adversarial attack results on RTTS [9]. The 1st and 3rd rows are the results of YOLOv3 [14] and the 2nd and 4th are those of Faster-RCNN [15].

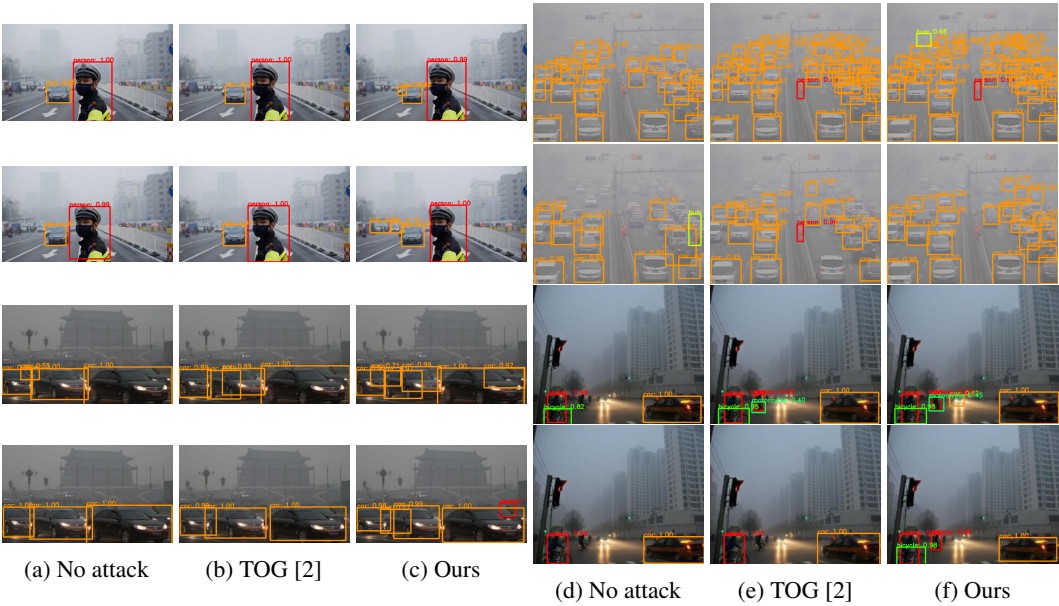

(a) No attack     (b) TOG [2]     (c) Ours     (d) No attack     (e) TOG [2]     (f) Ours

Figure 2: A comparison of object detection of different adversarial attack results on RTTS [9]. The 1st and 3rd rows are the results of YOLOv3 [14] and the 2nd and 4th are thoce of Faster-RCNN [15].

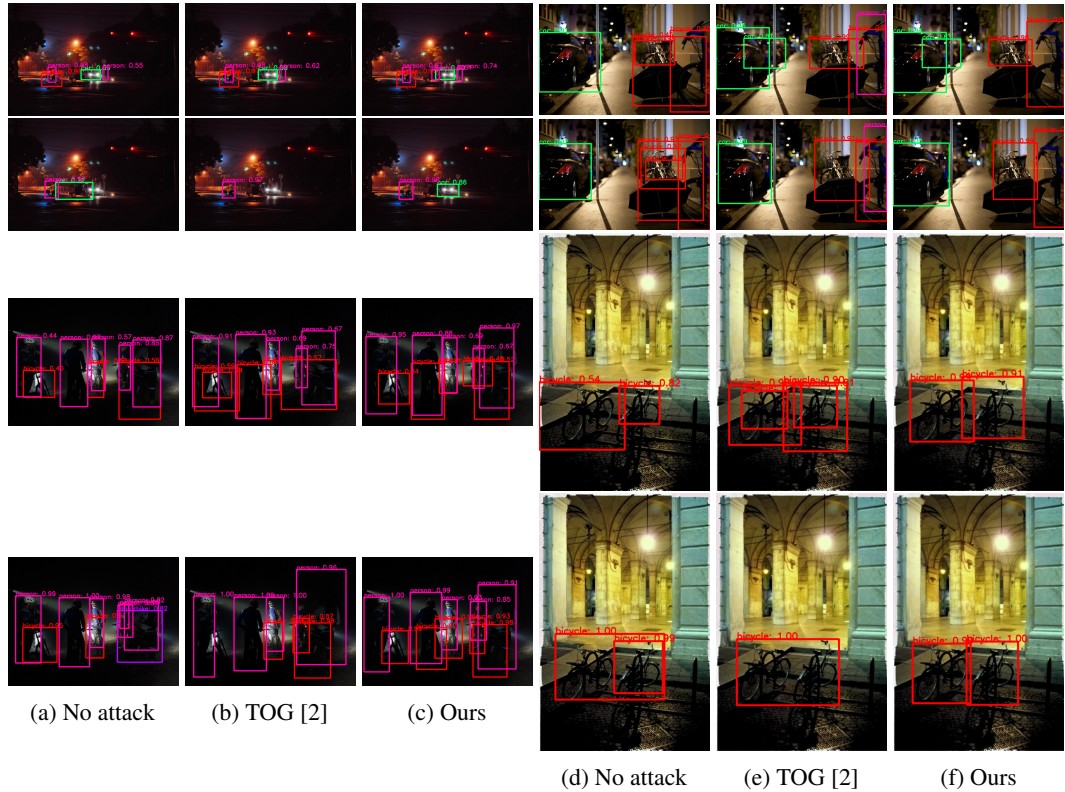

Figure 3: A comparison of object detection of different adversarial attack results on exdark [9]. The 1st and 3rd rows are the results of YOLOv3 [14] and the 2nd and 4th are those of Faster-RCNN [15].

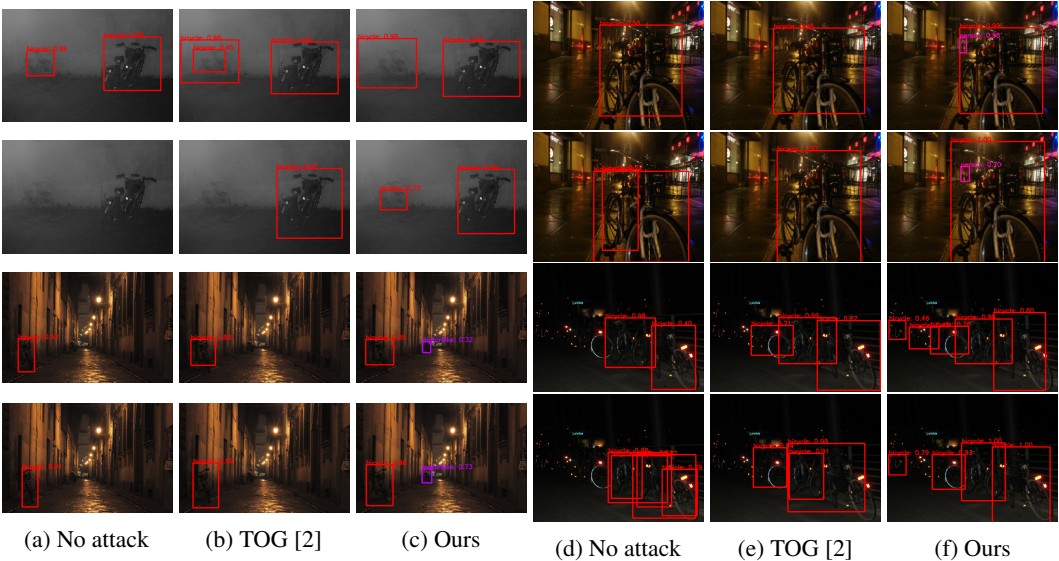

Figure 4: A comparison of object detection of different adversarial attack results on exdark [9]. The 1st and 3rd rows are the results of YOLOv3 [14] and the 2nd and 4th are those of Faster-RCNN [15].

# 4 More results on restoration for detection

The restoration and detection performance for Faster-RCNN [15] is shown ins Table 2 and Table 3.
Several more examples for detection and restoration performance are shown in Figure 5 8.

Table 2: The quantitative results of existing methods and ours with both restoration metrics and detection metrics on VOC_fog_test and RTTS. CT refers to conventional training. F denotes Faster-RCNN [15].

|  |  | F | MSBDN | | | GridDehaze | | |
|---|---|---|---|---|---|---|---|---|
|  |  | No restoration | F+CT | F+TOG | F+Ours | F+CT | F+TOG | F+Ours |
|  | mAP | 48.58 | 73.67 | 74.30 | 74.62 | 75.12 | 75.45 | 75.49 |
| \vocfogtest | PSNR | 13.50 | 28.72 | 27.78 | 27.87 | 27.42 | 27.07 | 27.04 |
|  | SSIM | 0.5411 | 0.8852 | 0.8728 | 0.8762 | 0.8667 | 0.8625 | 0.8634 |
| RTTS | mAP | 29.32 | 30.10 | 31.06 | 31.12 | 30.18 | 30.22 | 30.29 |

Table 3: The quantitative results of existing methods and ours with both restoration metrics and detection metrics on VOC_dark_test and ExDark. CT refers to conventional training. F denotes Faster-RCNN [15].

|  |  | F | SCI | | | ZeroDCE | | |
|---|---|---|---|---|---|---|---|---|
|  |  | No restoration | F+CT | F+TOG | F+Ours | F+CT | F+TOG | F+Ours |
|  | mAP | 56.37 | 59.89 | 60.12 | 60.21 | 59.77 | 60.03 | 60.06 |
| \vocdarktest | PSNR | 11.99 | 13.44 | 13.16 | 13.31 | 18.23 | 17.90 | 17.87 |
|  | SSIM | 0.3923 | 0.4891 | 0.4791 | 0.4803 | 0.6350 | 0.6362 | 0.6357 |
| ExDark | mAP | 39.91 | 42.20 | 42.38 | 42.34 | 40.47 | 41.18 | 41.32 |

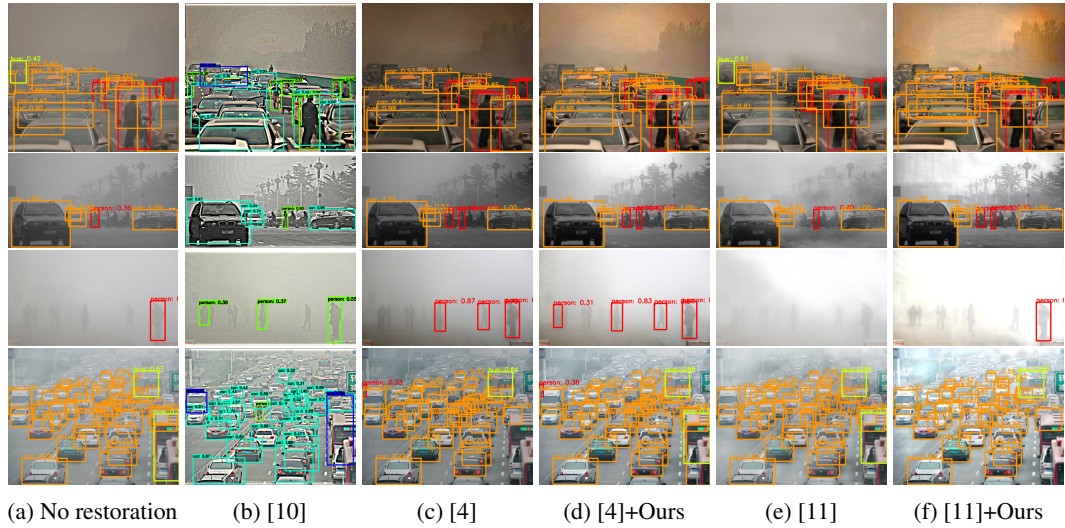

| (a) No restoration | (b) [10] | (c) [4] | (d) [4]+Ours | (e) [11] | (f) [11]+Ours |

Figure 5: A comparison of object detection of different adversarial attack results on RTTS [9]. The last four columns perform detection by YOLOv3[14].

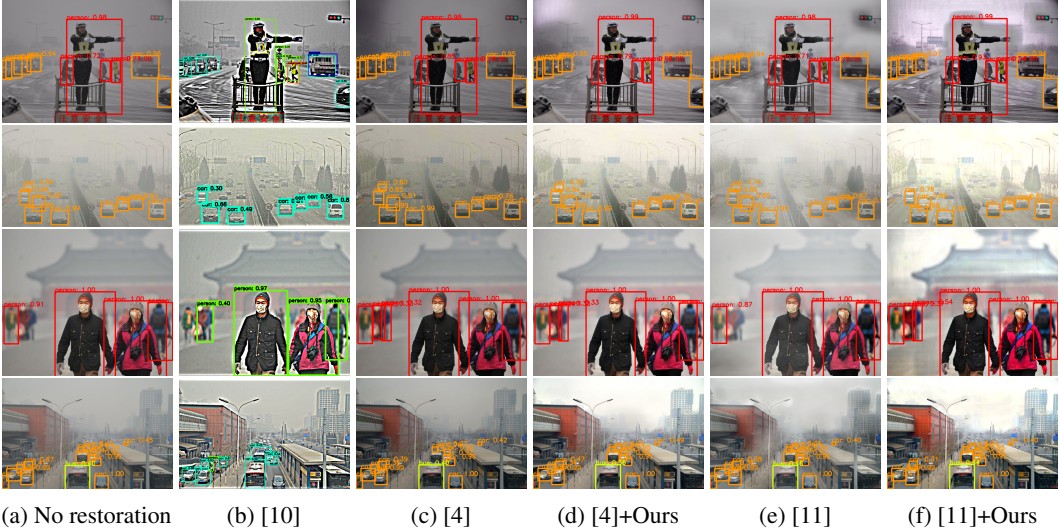

| (a) No restoration | (b) [10] | (c) [4] | (d) [4]+Ours | (e) [11] | (f) [11]+Ours |

Figure 6: A comparison of object detection of different adversarial attack results on RTTS [9]. The last four columns perform detection by Faster-RCNN [**?** ].

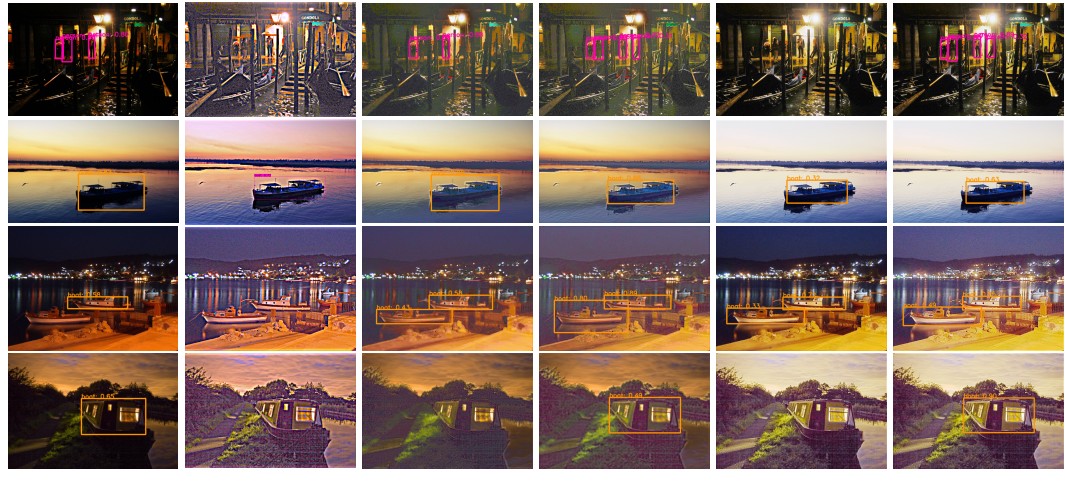

(a) No restoration      (b) [10]      (c) [6]      (d) [6]+Ours      (e) [13]      (f) [13]+Ours

Figure 7: A comparison of object detection of different adversarial attack results on ExDark [9]. The last four columns perform detection by YOLOv3[14].

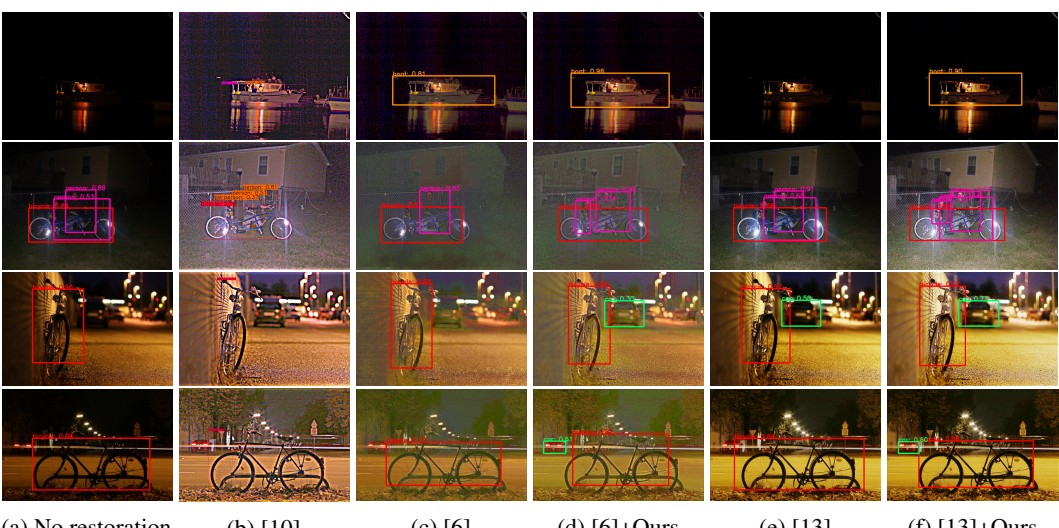

(a) No restoration      (b) [10]      (c) [6]      (d) [6]+Ours      (e) [13]      (f) [13]+Ours

Figure 8: A comparison of object detection of different adversarial attack results on ExDark [9]. The last four columns perform detection by Faster-RCNN[15].

## 5 Statistics of Datasets

The statistics of the used datasets are provided in Table 4 and Table 5. Since there are 5 categories in dehazing datasets and low light enhancement datasets respectively, we train YOLOv3 [14] and Faster-RCNN [15] with 5 classes on VOC_fog_train and 10 classes on vocdarktrain respectively.

Table 4: The statistics of VOC_fog_train [10], VOC_fog_test [10] and RTTS [9]

|  | image | person | bicycle | car | bus | motorcycle | total |
|---|---|---|---|---|---|---|---|
| VOC_fog_train | 8111 | 13256 | 1064 | 3267 | 822 | 1052 | 19561 |
| VOC_fog_test | 2734 | 4528 | 337 | 1201 | 213 | 325 | 6604 |
| RTTS | 4322 | 7950 | 534 | 18413 | 1838 | 862 | 29577 |

Table 5: The statistics of VOC_dark_train [10], VOC_dark_test [10] and ExDark [12]

|  | image | person | bicycle | car | bus | motorbike | boat | bottle | cat | chair | dog | total |
|---|---|---|---|---|---|---|---|---|---|---|---|---|
| VOC_dark_train | 12334 | 13256 | 1064 | 3267 | 822 | 1052 | 1140 | 1764 | 1593 | 3152 | 2025 | 29135 |
| VOC_dark_test | 3760 | 4528 | 337 | 1201 | 213 | 325 | 263 | 469 | 358 | 756 | 489 | 8939 |
| ExDark | 2563 | 2235 | 418 | 919 | 164 | 242 | 515 | 433 | 425 | 609 | 490 | 6450 |