# OpenReview forum: "Rethinking Image Restoration for Object Detection"
_NeurIPS.cc/2022/Conference — NeurIPS 2022 Accept_

### Official Review · Reviewer_owid · 2022-07-08

**Rating:** 8
**Confidence:** 5
**Soundness:** 3 good
**Presentation:** 3 good
**Contribution:** 4 excellent

**Summary:**

The paper rethinks image restoration for object detection from the perspective of adversarial examples and proposes a fine-tuning pipeline by formulating the tasks of image restoration and object detection into one. Unlike most existing methods that focus on modifying object detectors, the paper attempts to adapt restoration algorithms for generating high-quality visual perception and better detection results. To this end, a momentum-based ADAM-like iterative targeted adversarial example generation algorithm is designed to produce pseudo ground truth for fine-tuning restoration models. The proposed methods are evaluated on two image restoration tasks, i.e., image dehazing and low-light image enhancement, and extensive experimental results demonstrate its superiority to state-of-the-arts both in quality and quantity.

**Questions:**

See the weaknesses in the above.

**Limitations:**

The authors have addressed the limitations of the proposed method in the paper.

**Strengths And Weaknesses:**

Strengths:
1. Image restoration is usually viewed as a pre-processing for high-level computer vision tasks such as detection, but related discussions are rare in existing publications. The paper makes valuable explorations by formulating the two tasks into one.
2. Different from the conventional methods that modify object detectors, the proposed approach aims to adapt the restoration algorithms for improving the performances of both tasks.
3. The proposed Adam-variant adversarial example generation framework is based on Theorem 1, which is proved in the supplement, and thus becomes solid in theory.
4. Extensive experiments are conducted on various datasets, including a dehazing dataset and a low-light enhancement dataset, to demonstrate the effectiveness of the proposed method both in quality and quantity.
5. The paper is written well and easy to follow. It is good to illustrate the idea in Figure 1 and provide the proof and more analysis in the supplementary material.

Weaknesses:
1. L129 uses the math symbol D(x), while L131 utilizes D[R(x)]. It would be better to use a unified symbol. Similar typos are also present in equations 2, 3, and 5.
2. In the sentence “where \hat{x}^{*}=arg min_{x} ||\hat{x}^{*}-\hat{x}||” of equation 6, the symbol \hat{x}^{*} have two different meanings, i.e., a determined value on the left and a variable on the right.
3. In the captions of Tables 2 and 3, the word “ddetection” is a typo.
4. How to determine the initialization of the Number of attack iterations T, Update stepsize λ, and the Magnitude tolerance of Perturbation δ?

---

> ### Author Response · Authors · 2022-08-02
> **Response to Reviewer owid**
>
>
> ``Q-1:`` L129 uses the math symbol D(x), while L131 utilizes D[R(x)]. It would be better to use a unified symbol. Similar typos are also present in equations 2, 3, and 5.
>
> ``A-1:`` Thank you for kindly remind us of the typos. We will revise them.
>
> ``Q-2:`` In the sentence “where $\hat{x}^{}=arg min_{x} ||\hat{x}^{}-\hat{x}||$” of equation 6, the symbol $\hat{x}^{*}$ have two different meanings, i.e., a determined value on the left and a variable on the right.
>
> ``A-2:`` Thank you for kindly remind us of the typos. We will revise them.
>
> ``Q-3:`` In the captions of Tables 2 and 3, the word “ddetection” is a typo.
>
> ``A-3:`` Thank you for kindly remind us of the typos. We will revise them.
>
> ``Q-4:`` How to determine the initialization of the Number of attack iterations T, Update stepsize $\lambda$, and the Magnitude tolerance of Perturbation $\delta$?
>
> ``A-4:`` Thanks for the valuable question. We conduct a group of experiments as shown below about determining the coefficients of adversarial attack. The setting is elaborated in the caption. "PSNR" and "SSIM" are used to reflect how attacked images look different from original ones in terms of visual quality. "mAP" reflects how successfully images are attacked. We can find that $\delta$ heavily affect visual quality. A larger $\delta$ yields worse visual quality, but better detection. For a better visual quality, we choose $\delta=2/255$ where detection performance is sufficiently good. Regarding the pair of $\lambda$ and $T$, $\lambda=\frac{\delta}{T}$ is sufficient to generate good enough adversarial examples. A larger $T$ costs more time for adversarial attack. In our work, we thereby use $(\delta,\lambda,T)=(2/255,1/255,2)$.
>
>
> * The following table shows the experiment of selecting Number of attack iteration $T$, Update stepsize $\lambda$, Magnitude tolerance of Perturbation $\delta$ on a subset of VOC\_fog\_test with 100 images randomly selected. The object detector used for attack and subsequent detection is YOLOv3. The attack algorithm is ours. The adversarial examples are generated, on which "mAP" is computed, and "PSNR" and "SSIM" are computed between the examples and original images.
>
> |$\delta$|($\lambda$, $T$)|1/255,2|1/255,4|1/255,8|0.5/255,4|0.5/255,8|0.5/255,16|0.1/255,20|0.1/255,40|0.1/255,80|
> | :- | :- | :- | :- | :- | :- | :- | :- | :- | :- | :- |
> ||PSNR|49.16|47.72|47.40|49.21|47.60|47.39|49.31|48.02|48.49|
> |2/255|SSIM|0.9907|0.9869|0.9859|0.9908|0.9866|0.9859|0.9910|0.9879|0.9893|
> ||mAP(%)|94.50|94.62|94.59|94.04|94.46|95.54|95.92|96.78|96.87|
> ||PSNR|46.57|44.64|42.54|46.58|44.47|42.57|46.67|44.96|44.24|
> |4/255|SSIM|0.9831|0.9739|0.9582|0.9831|0.9728|0.9586|0.9835|0.9758|0.9717|
> ||mAP(%)|95.20|95.96|95.71|95.43|96.46|96.74|96.84|97.99|97.78|

---

### Official Review · Reviewer_2EoB · 2022-07-11

**Rating:** 4
**Confidence:** 3
**Soundness:** 2 fair
**Presentation:** 2 fair
**Contribution:** 2 fair

**Summary:**

Aiming at better object detection performance in bad environments such as hazy or low-light scenes, this paper proposes a new training protocol where an image restoration model is supervised by pseudo ground truths. The pseudo ground truths are generated using an adversarial attack method (Adam-like targeted attack in this paper). By doing so, the trained model shows better object detection performance in hazy or low light scenes than existing studies and plain models.

**Questions:**

Please see the Strengths And Weaknesses section.

**Limitations:**

Yes, the authors mention the limitations of their work but do not mention potential negative social impact.

**Strengths And Weaknesses:**

Originality

The proposed approach is interesting and it looks different from existing studies.

Quality

- Although the proposed approach is promising and interesting, my main concern is that the proposed method does not show good performance when image restoration models are trained on the pseudo samples of the training set and are evaluated on the test set. This seems natural because the difference between the images before/after performing the attack is slight, and thus it is difficult to learn.

- Another concern is that TOG achieves the almost same performance as the proposed method in Tables 2 and 3, indicating that the technical contribution of this work is limited.

- To show the effectiveness, it would be better to compare the proposed method with a simple baseline method that simultaneously optimizes the image restoration and object detector to minimize the loss of object detection.

Clarify

- There are several unclear points in this paper. For instance, what is "Y" in Tables 2 and 3? Also, what do you mean by "detection" in the caption of Table2? Finally, the authors mention that the mAP of YOLOv3 decreases by nearly 15% from that of clean images in lines 227 and 229, but where can we see the result of the clean images?

- It is somewhat unclear what is conducted and is the purpose of the experiment in Section 4.2.1. Please elaborate on it.

---

> ### Author Response · Authors · 2022-08-02
> **Response to Reviewer 2EoB**
>
>
> ``Q-1:`` Although the proposed approach is promising and interesting, my main concern is that the proposed method does not show good performance when image restoration models are trained on the pseudo samples of the training set and are evaluated on the test set. This seems natural because the difference between the images before/after performing the attack is slight, and thus it is difficult to learn.
>
> ``A-1:`` We agree that "the difference between the images before/after the attack is slight" and it's hard to learn. But we respectfully disagree that our method "does not show good performance" because it is challenging to improve detection performance and maintain good visual quality by only training a restoration network.
>
> In previous works, the performance gain of detection mainly comes from training detectors to adapt to adverse imaging environments. For example, the restoration subnetwork is trained together with a detector in both DSNet [15] and IA-YOLO [23]. However, traditional restoration models generate many artifacts as shown in Figure 6 of the manuscript. By removing the restoration subnetwork and simply training YOLOv3 on the hazy images of VOC\_fog\_train. The detection performance (we retrain a YOLOv3 that achieves 75.32\%) can easily exceed IA-YOLO (67.40\%) on VOC\_fog\_test.
>
> In summary, we can conclude that the performance gain comes from making the detector adapt to adverse or processed images. Our work is under a stricter and harder setting without changing or retraining the detector. As far as we know, we are the first to explore the problem and prove an effective way to address the problem. Our algorithm and pipeline are theoretically proven to work for all restoration methods for different types of degradation, and detectors of both single and two stages, which is valuable in the field. All the datasets in the work, e.g., RTTS and ExDark, are relatively large-scale, complicated, and naturally captured. Common performance gains on such benchmarks as well show great value.
>
> ``Q-2:`` Another concern is that TOG achieves the almost same performance as the proposed method in Tables 2 and 3, indicating that the technical contribution of this work is limited.
>
> ``A-2:`` Our main contribution mainly lies in introducing adversarial attacks to generate pseudo ground truth for better downstream detection and restoration quality. Both TOG and ours can generate compelling pseudo ground truth, which shows the effect of our proposed fine-tuning pipeline. Furthermore, the proposed algorithm can theoretically and experimentally outperform TOG for most restoration methods and detectors of both single and two stages. This shows a valuable contribution.
>
> ``Q-3:`` To show the effectiveness, it would be better to compare the proposed method with a simple baseline method that simultaneously optimizes the image restoration and object detector to minimize the loss of object detection.
>
> ``A-3:`` To show the effectiveness of the proposed method, we compare the proposed method against baseline methods (DSNet [15] and IA-YOLO [23]) that simultaneously optimize the image restoration and object detector to minimize the loss of object detection. However, as shown in Tables 2 and 3 in the manuscript,  DSNet [15] and IA-YOLO [23] yield worse visual quality and detection accuracy results than our proposed methods.
>
> In addition, we also conduct an experiment to optimize the restoration model by summing restoration loss and detection loss and keeping the detector unchanged. The results of MSBDN on VOC\_fog\_test with different loss weights are shown in the following table. We can find the introduction of detection loss consistently decreases PSNR and SSIM by creating artifacts. However, the mAP cannot exceed our proposed pipelines (77.52\% of TOG and 77.66\% of ours). Therefore, the detection loss cannot propagate to restoration efficiently without changing the detector's parameters.
>
> * The table below shows the experiment on optimizing MSBDN on VOC\_fog\_test by weighted summation of restoration loss and detection loss, i.e. $L_{restoration}+\gamma L_{detection}$. The detection model is remained unchanged during fine-tuning.
>
> |$\gamma$|0.1|1|4|5|6|10|
> | - | - | - | - | - | - | - |
> |PSNR|28.64|26.92|26.58|26.55|26.56|24.68|
> |SSIM|0.8848|0.8679|0.8667|0.8654|0.8642|0.8424|
> |mAP(%)|77.09|77.12|77.26|77.37|77.22|77.14|

---

> > ### Comment · Reviewer_2EoB · 2022-08-09
> > **Response to the authors**
> >
> > Thank you so much for the response.
> >
> > Although I've read the feedback from the authors, my main concern remains that the proposed method's performance gain is limited on the test set. For instance, Table 2 shows only 0.14 and 0.08 mAP improvements on VOC_fog_test and RTTS from TOG for YOLOv3+MSBDN. Moreover, there are 0.43 and 0.04 improvements on VOC_fog_test and RTTS for YOLOv3+GridDehaze. We can also say the same thing for the result in Table 3 and PSNR/SSIM values.
> > These facts do not indicate that the proposed method is sufficiently effective.
> >
> > Thus, the authors mention in the response that "we are the first to explore the problem and prove an effective way to address the problem. Our algorithm and pipeline are theoretically proven to work for all restoration methods for different types of degradation, and detectors of both single and two stages", but this statement seems overclaimed.
> >
> > I agree that the proposed approach is interesting and promising, but its effectiveness is not supported well by the experiments and their results.

---

> > > ### Author Response · Authors · 2022-08-09
> > > **Response to Reviewer 2EoB**
> > >
> > > Thank you for your issue. But we still want to emphasize the universality of our method. Our algorithm can extend to most of the restoration networks and detection networks. For most restoration tasks, e.g., haze removal (Table 2), low-light enhancement (Table 3), and the case with multiple degradations (the table responded to Reviewer 6L2X), fine-tuning the restoration network can improve the performance of detection networks. Especially when multiple kinds of degradation exist, the improvement of mAP achieves better results by up to 1% as shown in the table below. We also admit that our method can be improved in terms of performance, therefore, we will focus on better learning the difference between the images before/after performing the attack from pseudo-ground truth as suggested.
> > >
> > > * The table below shows the experiment when two kinds of degradation exist, i.e. low light and haze. The restoration and detection methods are NDNet [Reference 2] and YOLOv3 respectively.
> > >
> > >
> > > ||no restoration|Conventional Training|TOG|Ours|
> > > | :- | :- | :- | :- | :- |
> > > |PSNR|11.92|25.53|25.35|25.34|
> > > |SSIM|0.5375|0.8603|0.8564|0.8576|
> > > |mAP(%)|53.01|69.55|70.11|70.50|

---

> ### Author Response · Authors · 2022-08-02
> **Response to Reviewer 2EoB (Continued)**
>
>
> ``Q-4:`` There are several unclear points in this paper. For instance, what is "Y" in Tables 2 and 3? Also, what do you mean by "detection" in the caption of Table2? Finally, the authors mention that the mAP of YOLOv3 decreases by nearly 15\% from that of clean images in lines 227 and 229, but where can we see the result of the clean images?
>
> ``A-4:`` Thank you for the request of clarification. "Y" in Tables 2 and 3 means YOLOv3. A single "Y" means to directly perform detection by YOLOv3. We will clarify this in the revised paper.
>
> The mentioned mAPs of YOLOv3 are 81.91\%-66.88\%=15.03\% for VOC\_fog\_test, which means YOLOv3's performance decreases by 15.03\% before and after adding synthetic haze on the clean images of VOC\_fog\_test. Similar case for 14.11\% on VOC\_dark\_test.
>
> We will carefully proofread the manuscript and revise all the typos in the revised paper.
>
> ``Q-5:`` It is somewhat unclear what is conducted and is the purpose of the experiment in Section 4.2.1. Please elaborate on it.
>
> ``A-5:`` Section 4.2.1 evaluates whether adversarial perturbation can boost detection performance in common circumstances. Our experiment is based on the following hypothesis: only if the adversarial perturbation improves detection performance can it be used for fine-tuning restoration networks.
>
> Therefore, we conduct experiments to firstly show that TOG can make adversarial examples on which detection accuracy increases (e.g., from 42.77\% to 71.98\% on RTTS) only with a very small perturbation added. Compared to TOG, our ADAM-like algorithm further increases mAP (to 78.50\%). So intuitively, dehazing model learning from pseudo ground truth generated by ours can recover images with better downstream detection performance, which is demonstrated in Sections 4.2.2 and 4.2.3 in the manuscript.

---

### Official Review · Reviewer_6L2X · 2022-07-11

**Rating:** 5
**Confidence:** 4
**Soundness:** 3 good
**Presentation:** 3 good
**Contribution:** 3 good

**Summary:**

the proposes a fine-tuning approach to improve the object detection performance without reducing the visual quality of the restored image. To achieve the authors follow a target adversarial attack on the object detection task  in order to improve the detection performance and on the other hand they don't effect the restoration task optimization.

**Questions:**

Please refer weaknesses

**Ethics Review Area:**

["I don’t know"]

**Limitations:**

Authors discussed limitations that are relevant to the proposed method

**Strengths And Weaknesses:**

Strengths:
- the paper proposes a simple technique that improves the object detection performance and on the other hand doesn't decrease the image restoration performance.
- they generate adversarial samples using the second-order gradients and cumulative momentum approach.

Weaknesses:
- Why is the performance using the proposed method significantly less in table 2 and table 3.
-  Can authors show zoomed versions of visualizations for easy understanding
- Can authors explain while generating adversarial attack samples how are they maintain image properties like how are the authors make sure whether the adversarial attack image naturally possible image or not.
- what happens if the images have multiple degradations will the proposed method still applicable

---

> ### Author Response · Authors · 2022-08-02
> **Response to Reviewer 6L2X**
>
>
> ``Q-1:`` Why is the performance using the proposed method significantly less in table 2 and table 3.
>
> ``A-1:`` In Table 2-3, our method achieves better results than the conventional training strategy. For example, compared with the original training data, the detection performance improvement of YOLOv3 is significant (77.66\% vs. 77.06\% in terms of mAP) using the proposed pseudo ground truth to train the MSBDN model on the VOC\_fog benchmark as shown in Table 2. Furthermore, our proposed attack algorithm also yields better pseudo ground truth than the recent attack model of TOG, e.g., 77.66\% vs. 77.52\% in Table 2.
>
> Compared to Table 1, in Tables 2 and 3, training or fine-tuning of restoration method is conducted to learn the attacked pseudo ground truth. The learning process is hard since the perturbation is small. However, in Table 1, only adversarial attack is conducted to evaluate whether adversarial perturbation can boost detection performance in common circumstances. Therefore, the detection performance in Table 1 is much higher than Tables 2 and 3 where restoration method is trained or fine-tuned to learn the small perturbation.
>
> ``Q-2:`` Can authors show zoomed versions of visualizations for easy understanding
>
> ``A-2:`` Thank you for the valuable suggestion. We will revise the paper with zoomed figures for better visualization.
>
> ``Q-3:`` Can authors explain while generating adversarial attack samples how are they maintain image properties like how are the authors make sure whether the adversarial attack image naturally possible image or not.
>
> ``A-3`` The clipping operation in Eq. 10 regulates the perturbation to be small enough for human eyes. "PSNR" and "SSIM" are commonly used to represent the closeness between a pair of images. We compute the PSNR and SSIM on six datasets. When PSNR exceeds 45 and SSIM exceeds 0.99 between two images, the pixel-wise difference is very slight and negligible [Referece 1]. From the perspective of human vision, the difference can hardly be recognized. As shown in the following tables, all the average PSNR and SSIM exceed 48 and 0.98 respectively. Such a small perturbation cannot change the natural property of attacked images. In summary, the clipping operation ensures the perturbation is sufficiently small so that natural image properties are maintained.
>
> * The following six tables show the detection performance gain by different targeted adversarial attack methods on YOLOv3. The setting is $\delta=2/255$, $\lambda=1/255$ and $T=2$. The average PSNR and SSIM between the natural original images and their corresponding adversarial examples are computed for six datasets.
>
> * RTTS
>
> ||no attack|TOG|Ours|
> | :- | :- | :- | :- |
> |PSNR|Inf|48.54|48.97
> |SSIM|1|0.9901|0.9905
> |mAP(%)|42.77|71.98|78.50
>
> * Hazy images of VOC\_fog\_test
>
> ||no attack|TOG|Ours|
> | :- | :- | :- | :- |
> |PSNR|Inf|49.21|49.12
> |SSIM|1|0.9821|0.9898
> |mAP(%)|66.88|92.15|94.12
>
> * Clean images of VOC\_fog\_test
>
> ||no attack|TOG|Ours|
> | :- | :- | :- | :- |
> |PSNR|Inf|49.30|49.24
> |SSIM|1|0.9952|0.9938
> |mAP(%)|81.91|94.81|96.28
>
> * ExDark
>
> ||no attack|TOG|Ours|
> | :- | :- | :- | :- |
> |PSNR|Inf|48.43|48.49
> |SSIM|1|0.9919|0.9916
> |mAP(%)|46.27|74.29|79.29
>
> * Dark images of VOC\_dark\_test
>
> ||no attack|TOG|Ours|
> | :- | :- | :- | :- |
> |PSNR|Inf|48.35|48.47
> |SSIM|1|0.9937|0.9957
> |mAP(%)|56.88|75.69|81.99
>
> * Clean images of VOC\_dark\_test
>
> ||no attack|TOG|Ours|
> | :- | :- | :- | :- |
> |PSNR|Inf|49.18|49.04
> |SSIM|1|0.9940|0.9933
> |mAP(%)|70.99|84.68|88.83

---

> ### Author Response · Authors · 2022-08-03
> **Response to Reviewer 6L2X (Continued)**
>
>
> ``Q-4:`` What happens if the images have multiple degradations will the proposed method still applicable
>
> ``A-4:`` To evaluate the proposed algorithm on multiple degradations, we train and test a nighttime dehazing algorithm NDNet [Reference 2], which can simultaneously remove haze and brighten images. For training and testing, we simulate a dark foggy version of VOC, dubbed VOC\_fog\_dark, including the VOC\_fog\_dark\_train set and VOC\_fog\_dark\_test set. Then, we experiment on the new dataset and the results are shown below. Our method is applicable when two kinds of degradation exist.
>
> * The table below shows the experiment when two kinds of degradation exist, i.e. low light and haze. The restoration and detection methods are NDNet [Reference 2] and YOLOv3 respectively.
>
>
> ||no restoration|Conventional Training|TOG|Ours|
> | :- | :- | :- | :- | :- |
> |PSNR|11.92|25.53|25.35|25.34|
> |SSIM|0.5375|0.8603|0.8564|0.8576|
> |mAP(%)|53.01|69.55|70.11|70.50|
>
> In addition, if we consider noise as a natural degradation, most of the existing dehazing and low-light enhancement algorithms can be regarded as handling multiple degradations (e.g., haze & noise, low-light & noise) in one go. Therefore, if we consider image dehazing and low-light enhancement in this way, our proposed method could be seen as applicable for multiple degradations.
>
> [Reference 1] De Rosal Igantius Moses Setiadi. 2021. PSNR vs SSIM: imperceptibility quality assessment for image steganography. Multimedia Tools Appl. 80, 6 (Mar 2021), 8423–8444. https://doi.org/10.1007/s11042-020-10035-z
>
> [Reference 2] Jing Zhang, Yang Cao, Zheng-Jun Zha, and Dacheng Tao. 2020. Nighttime Dehazing with a Synthetic Benchmark. In Proceedings of the 28th ACM International Conference on Multimedia (MM '20). Association for Computing Machinery, New York, NY, USA, 2355–2363. https://doi.org/10.1145/3394171.3413763

---

### Author Response · Authors · 2022-08-08
**Manuscript is revised**

Dear Reviewers, the revised manuscript is uploaded. If you have any other concern, please feel free to raise it. Thank you sincerely for your effort and attention.

---

### Meta-Review · Area_Chair_FiCA · 2022-08-29

**Recommendation:** Accept
**Confidence:** Certain

**Metareview:**

In this paper, the authors provide an interesting formulation of an adversarial attack that can directly help object detector training in the presence of various degradations. This is a departure from the usual formulation of restoration, followed by detector training. I liked the initial derivation which is elegant and logical, and their experimental results show that mAP is clearly improved over baseline training. 2 of the 3 reviewers supported acceptance (with one being a strong accept). The third reviewer felt that the experimental improvements were insufficient. While I agree that the mAP improvements are not in the "wow" category, I still think the method is solid (as also accepted by third reviewer) and worthy of acceptance.

**Award:**

No

---

### Decision · Program_Chairs · 2022-09-14

Accept